# OpenReview forum: "WASP: Benchmarking Web Agent Security Against Prompt Injection Attacks"
_NeurIPS.cc/2025/Datasets_and_Benchmarks_Track — NeurIPS 2025 Datasets and Benchmarks Track poster_

### Official Review · Reviewer_9Vtm · 2025-06-27

**Rating:** 4
**Confidence:** 4

**Summary:**

This paper introduces WASP, which is a publicly available benchmark for end-to-end evaluation of Web Agent Security against prompt injection attacks. WASP is a dynamic benchmark built within a sandbox web environment based on the existing work (VisualWebArena). The paper simulates prompt injection attacks in web environments. The paper also conducts evaluations and provides the corresponding analysis for the obtained evaluation results.

**Additional Feedback:**

[-] There is a typo in Line 307, i.e., “First, table 3 examines….”.

**Dataset Code Accessibility:**

Partly

**Ethical Considerations:**

No, there are no or only very minor ethics concerns

**Final Justification:**

I have carefully read the responses. My concerns are addressed.

**Limitations Weaknesses:**

[-] The threat model considered in the paper is unclear and lacks sufficient detail. First, in the second paragraph of Section 2, the paper does not clearly specify how many parties are involved in the threat model. Additionally, the communication and interactions among these parties are not well explained, making it difficult to fully understand the attack scenarios. Are there any differences among the considered attack scenarios in terms of threat models? This issue also extends to the defense section of the proposed benchmark, where the number of parties and their roles remain ambiguous. To improve clarity, it would be helpful for the authors to include a detailed diagram illustrating the workflow and information flow among the entities involved in the threat model.

[-] The paper does not discuss whether existing related works can be generalized to realistic settings. It remains unclear what specific challenges prevent such generalization. Additionally, in the second paragraph of Section 1, the paper only provides the high-level descriptions of existing prior works, and lacks a clear comparison between its settings and those used in prior works. Providing a detailed comparison in a table would help clarify how the proposed benchmark differs from or improves upon existing methods in terms of realism and applicability to real-world scenarios.

[-] The paper considers multiple attack steps. However, the paper does not clearly explain how these differ from single-step attacks in terms of effectiveness or attack dynamics. It would be helpful to clarify what unique advantages or challenges multi-step attacks introduce compared to single-step attacks. Additionally, the paper lacks an analysis of the computational cost associated with these attacks under different evaluation settings. Moreover, the evaluation includes human-written injections, but the paper does not report the time or effort required to generate them.

[-] The descriptions for the attacker’s capabilities and roles are confusing. For example, in Line 84, the paper describes an attack scenario where the agent’s user is benign, whereas in Line 86, it states that the attacker is an adversarial user of a website. This creates confusion about who the attacker is, what their access privileges are, and how they interact with the system. Furthermore, in Line 124, the authors state that the paper considers only black-box attackers, yet in Line 138, the attacker is described as having control over specific webpage elements, which implies some level of system knowledge or access. Moreover, in Line 167, the paper assumes the adversary has full knowledge of the user goal. These inconsistencies make it difficult to clearly understand the attacker’s capabilities. It would be helpful if the authors could explicitly outline their knowledge, capabilities, and control surfaces. Including these information in the attack and defense workflows would also improve clarity.

[-] The paper explores defenses against the studied prompt injection attacks; however, it lacks a discussion of existing or potential defense strategies, which makes difficult to understand defenses within the broader landscape of prior work. Furthermore, the benchmark evaluation uses only two existing defenses, which limits the comprehensiveness of the evaluation. It is also unclear why these two particular defenses were selected and why other relevant defenses were excluded. A more thorough justification for the defense choices, along with broader inclusion of existing methods, would strengthen the comprehensiveness and credibility of the proposed benchmark.

[-] The evaluation conducted for the proposed benchmark is insufficient and focuses on a limited range of scenarios. For example, while the paper explores attack settings where the environment is malicious and the agent’s user is benign, it only includes two environments, which limits the diversity and generalizability of the results. Additionally, the benchmark only considers two types of prompt injection attacks. It is unclear how many distinct types of prompt injection attacks have been identified in existing literature, and whether the proposed benchmark can support the implementation of additional attack variants.

**Strengths Contributions:**

[+] The paper tests prompt injection attacks as well as task performance in an end-to-end manner in an isolated and controllable environment.

[+] The paper conducts evaluations for the proposed benchmark, and provides findings and analysis for the obtained evaluation results.

[+] The code and benchmark are publicly available.

---

> ### Author Rebuttal · Authors · 2025-07-30
>
> Thank you to the reviewer for the detailed review! Here are our responses:
>
> > [-] The threat model considered in the paper is unclear and lacks sufficient detail. (...) it would be helpful for the authors to include a detailed diagram illustrating the workflow and information flow among the entities involved in the threat model.*
>
> **Response**: Our paper considers three main parties: a benign user of the agent, an autonomous web agent acting on that user’s behalf, and an adversarial user of the websites who injects malicious content into pages the agent visits. The agent interacts with the environment to fulfill its user’s goals, while the adversary operates within realistic constraints—only able to manipulate elements typically accessible to regular users of the web page (e.g., comments, posts). We acknowledge that a diagram could further clarify these interactions and will include one in future revisions. The attack scenarios all follow this core model, with variations only in the specific attacker goals and text, not in the fundamental roles or communication patterns.
>
> > [-] The paper does not discuss whether existing related works can be generalized to realistic settings. (shortened due to rebuttal word limit).*
>
> **Response**: Our paper addresses the gap in realism by evaluating agents in fully operational, self-hosted web environments, whereas many existing benchmarks rely on simulated or overly simplified settings. We provide a detailed comparison in table 1, such as limited attacker privileges, multi-step workflows, and end-to-end evaluation—and we will add more discussion in the next revision. We also provide a detailed comparison to prior work in section 2 (Background), between lines 79 and 125).
>
> We would also like to further clarify our distinct additional contribution on top of AgentDojo in particular. AgentDojo operates in an environment with simulated LLM responses to tool calls and does not include web agents (only 4 APIs). By contrast, our benchmark is built on fully operational, self-hosted web platforms seeded with real data that closely mirror real-world usage and threats. The security of web and API agents is inherently different: While agents relying on APIs are restricted by the design of the API, web agents come with much more permission by default and no fine-grained access controls. For example, functions to manage the keys used to access a GitLab repository are not be available to an API agent such as those in AgentDojo but web agents by default come with the full permissions of a logged-in user of a website and can take more destructive/impactful actions. Additionally, adversaries face different restrictions in API and in web agents. Some web agents operate only with a screenshot view and do not take in a text representation of the page. This means adversaries are limited in the length of their injected prompt. By contrast, an email with an injection in AgentDojo can be arbitrarily long. We believe it is important for the community to be able to study attacks on realistic agentic workflows both on the web and in an API setting. That is why WASP intends to set a level playing field for attackers and modeling/defender teams under standardized and realistic threat models on the web. By focusing on realistic web environments, we are able to evaluate web agents under authentic, multi-step workflows and adversarial conditions that agents are likely to encounter in practice
>
>
> > [-] (...) the paper does not clearly explain how [multi-step attacks] differ from single-step attacks in terms of effectiveness or attack dynamics. (shortened due to rebuttal word limits).*
>
> **Response**: Multi-step attacks in our benchmark more closely mirror real-world adversarial scenarios, requiring agents to navigate complex workflows. This approach exposes vulnerabilities that single-step attacks may miss. Our empirical findings back the need for this kind of multi-step benchmark: agents struggle with completing the end-to-end tasks of moderate complexity. Consider the Utility column in Table 2: almost none of the tested agents can complete more than half of tasks that the legitimate user of the agent requests with no adversary present. Additionally, we hand-labeled some trajectories of agents under attack to better understand attack failure modes. Those results are shown in Figure 3. Note that in most cases an agent does not recover after being hijacked but rather simply fails to complete the task set by the adversary (the red set of states on the left flows to mostly red states to the right). This points to the fact that our attacks are successful at getting the agents to adopt the adversarial goals but the utility of the agents to complete them is limited.
>
>
> These results show that many agents are only “safe” because they fail to fully execute complex tasks, a phenomenon we describe as “security through incompetence.” If an agent is consistently diverted from its original objective by adversarial prompts, it signals that the agent is susceptible to manipulation, and as agent capabilities improve, these currently “safe” failures could easily become real security incidents. By identifying when the agent’s reasoning is compromised, we can proactively address weaknesses before they result in harmful or unsafe behaviors as agents become more competent and autonomous.
>
> Regarding effort, we were able to discover the templates for the attacks within half a day of manual interactions with the agents. We are happy to include this discussion in the paper. We do not conduct computational attacks, so the point of the reviewer is well taken but we would like to point out that no computational attacks are included in our work.
>
>
> > [-] The descriptions for the attacker’s capabilities and roles are confusing. (...).*
>
> **Response**: Thank you for bringing this up and we would be happy to clarify the text of the paper! In the meantime, we would like to point out that the quoted text itself suggests those are two different kinds of users: the user of the agent (who is benign) and the user of a website the agent visits (who is adversarial). In our threat model, the attacker is always an adversarial user of the web platform, with no special system privileges—only the ability to inject content where any user could (e.g., comments, posts). The agent’s user is always benign. The “black-box” designation means the attacker has no knowledge of the agent’s internal workings, but may know the user’s goal if it is visible in the environment. We will revise the text to make these distinctions explicit and ensure consistency throughout, and will consider adding a summary table of attacker knowledge, capabilities, and control surfaces for further clarity.
>
>
> > [-] The paper explores defenses against the studied prompt injection attacks; however, it lacks a discussion of existing or potential defense strategies, which makes difficult to understand defenses within the broader landscape of prior work. (...).*
>
> **Response**: Thank you for raising the point about defense coverage. Our benchmark evaluates the two most popular prompt injection defenses in realistic web agent settings: system prompt hardening and instruction hierarchy. We picked those two because they are representative of current best practices and are supported by the agentic frameworks we evaluated. However, our benchmark is modular and open-source, so it is easy to extend to test different defenses and we invite the community to do so. Our main contribution is on establishing better defined attackers (adversarial users of websites the agent visits) and realistic multi-step adversarial goals and we only provide a set of baselines of attacks and defenses that we hope the community will build upon within this framework.
>
> > [-] The evaluation conducted for the proposed benchmark is insufficient and focuses on a limited range of scenarios. (...).*
>
> **Response**: Thank you for your feedback on the evaluation scope. We agree that broader coverage is important, and we want to clarify our reasoning and context for the current setup:
> - The two environments we selected allow us to test a wide range of attacker and user goals, and to test multi-step, end-to-end workflows in depth. In addition, we already include adversarial objectives spanning from security (exceeding authorized access) to social harms (harassment, amplifying political messages, etc). Our results are consistent across these scenarios in two different environments. The vulnerabilities and attack patterns we observe are not unique to GitLab or Reddit—they’re likely to show up in any environment where agents interact with untrusted user content.
> - Running large LLMs—especially agentic models—is extremely resource-intensive. The benchmarking community is moving toward smaller but more diverse benchmarks (see arXiv:2402.14992). We followed this trend to keep the time and cost manageable: for example, running Claude-cua on our full benchmark takes about 72 hours and costs ~$1,500. So scaling this up to thousands of instances would be introducing a lot of extra cost with marginal benefit to generalizability.
> - Our benchmark size is 84 test cases, which is actually on par with several recent agentic benchmarks: AgentHarm (110 tasks), Tau-bench (165 tasks), etc. So, while the number of environments is limited, the overall task diversity and coverage are quite competitive.
> - The WASP framework is open-source and modular, so it’s straightforward to add new environments like e-commerce or travel booking in the future. We explicitly mention this as a direction for future work and welcome community contributions.
> In short, we focused on depth and realism within two environments to keep the benchmark practical and meaningful, while still offering broad task coverage. Expanding to more environments is definitely on our roadmap, and we see this as a community effort going forward.

---

> > ### Comment · Reviewer_9Vtm · 2025-08-05
> > **Thank you for your responses**
> >
> > Thank you for your responses. My concerns are addressed. I will raise my score.

---

> > > ### Author Response · Authors · 2025-08-06
> > >
> > > Thank you! We appreciate the thoughtful engagement!

---

### Official Review · Reviewer_RYwN · 2025-06-29

**Rating:** 5
**Confidence:** 4

**Summary:**

This paper introduces WASP (Web Agent Security against Prompt injection), a new benchmark for evaluating the security of autonomous web agents against prompt injection attacks. The authors argue that existing benchmarks are either too simplistic or grant attackers unrealistic capabilities. WASP aims for greater realism by using a sandboxed web environment (based on VisualWebArena with gitlab and reddit clones) where attackers have limited, realistic control—they can only inject content where normal users can, like in comments or posts. The paper evaluates several SOTA AI models and agentic frameworks, including GPT-4o, Claude 3.5 Sonnet, and OpenAI's o1 model. The primary finding is that while agents are easily hijacked (high intermediate attack success rate, up to 86%), they often fail to complete the attacker's full objective due to current capability limitations, a phenomenon the authors term "security by incompetence."

**Additional Feedback:**

This is a strong and timely paper that introduces a valuable asset to the community. The "security by incompetence" finding is particularly insightful. Here are some additional points and questions for the authors:

1. Clarity - there are minor formatting inconsistencies in the paper. For example, references to figures and tables switch between capitalized (e.g., "Table 1," "Figure 2") and lowercase (e.g., "fig. 3," "table 5"). Please ensure consistent formatting throughout. Additionally, some underlines (e.g., under "e.g." on L326) seem unnecessary and could be removed for better readability.

2. In Table 2, the ASR rates are reported for a mix of URL and plain-text injections. However, Table 3 splits the results by injection type. To improve clarity, please either specify the ratio of URL to plain-text injections used for the aggregate results in Table 2 or consider merging the tables.

3. Analysis of Agent Reversion: Figure 3 shows some fascinating cases where the agent reverts to the original user's request after being hijacked. Could the authors investigate this further? It would be valuable to understand if this reversion is a sign of a latent security awareness (i.e., the agent realizes it was misled) or simply another form of incompetence where it fails to follow the attacker's instructions correctly.

**Dataset Code Accessibility:**

Yes

**Dataset Code Comments:**

The authors state that the benchmark code is open-sourced and publicly available. The NeurIPS checklist in the paper provides a GitHub link (https://github.com/facebookresearch/wasp), fulfilling the requirement for accessibility. The paper provides sufficient detail on the setup (AWS EC2 instances, model APIs used) and scaffolding to support reproducibility, as described in Section 4.1.

**Ethical Comments:**

- All experiments are conducted within a sandboxed, self-hosted environment.
- The research does not involve human subjects.

**Ethical Considerations:**

No, there are no or only very minor ethics concerns

**Final Justification:**

Keeping my current score of 5. This has included the rebuttal discussion with the authors. Overall I was happy with the paper at the beginning and happier after the authors addressed my issues, but I don't feel like it's enough for a Strong Accept, thus keeping the score at Accept.

**Limitations Weaknesses:**

1. The authors acknowledge the benchmark is limited to two environments (gitlab and reddit). However, this limitation extends to the tasks themselves, with only two user goals per environment. This raises questions about the benchmark's diversity and ability to cover a wide range of scenarios. The benchmark would be strengthened by incorporating more tasks, including more complex ones that agents currently fail (as seen in the utility test), to better quantify the degree of "security by incompetence" and to "future-proof" the benchmark against more capable future agents.

2. The paper does not categorize the attacks based on the type of malicious behavior being attempted. For instance, attacks could be categorized as information theft, permission abuse (e.g., an agent using the user's credentials to perform unauthorized actions), or causing financial harm (e.g., making incorrect purchases). A taxonomy of attacks would be valuable for identifying which specific agent functionalities are most vulnerable and for designing more granular security controls.

3. The core finding of "security by incompetence" highlights that the low end-to-end success rate is primarily a function of the agent's general capability, not its security robustness. While interesting, this means the security-specific insights are somewhat limited. The benchmark could be improved by designing tasks that better isolate security mechanisms from general task execution competence.

4. Some analyses in the paper could be deeper. For example, the o1 model had a surprisingly high ASR-intermediate despite its instruction hierarchy. The authors' initial analysis is appreciated, but further experiments would be good to root-cause this vulnerability and potentially flag a specific loophole in o1's security design.

**Strengths Contributions:**

1. The primary contribution is the WASP benchmark itself. It addresses a critical gap in agent security evaluation by creating a more realistic threat model than prior work. Unlike benchmarks that assume the attacker has full control of the environment, WASP models attackers as adversarial users with limited capabilities (e.g., posting comments), which more accurately reflects real-world scenarios on trusted websites. The distinction from prior work like InjecAgent and ASB is clearly articulated in Table 1 and the "Comparison with Prior Work" section.

2. By measuring both the initial hijacking of the agent (ASR-intermediate) and the final completion of the attacker's goal (ASR-end-to-end), the authors uncover a crucial insight: current agents, even when successfully compromised, often fail to execute the malicious multi-step task. This concept of "security by incompetence" is a novel and important observation for the field, suggesting that as agent capabilities improve, their security risks will become more severe.

3. The authors test a strong and timely set of modern, powerful models (GPT-4o series, Claude Sonnet 3.5/3.7) and agentic frameworks (VisualWebArena, Claude CURI, Tool Calling). This makes the results highly relevant and impactful for researchers and developers currently working on these systems.

4. Clarity and Organization: The paper is well-written, clearly structured, and easy to follow. The threat model, experimental setup, and metrics are well-defined. Figures and tables, such as Figure 1 (results summary and example), Figure 2 (attack illustration), and Table 2 (main results), are informative and effectively support the paper's claims.

---

> ### Author Rebuttal · Authors · 2025-07-30
>
> Thank you to the reviewer for acknowledging the contributions and providing feedback that will make this paper stronger! Here are our responses to each point:
>
> > The authors acknowledge the benchmark is limited to two environments (gitlab and reddit). However, this limitation extends to the tasks themselves, with only two user goals per environment. This raises questions about the benchmark's diversity and ability to cover a wide range of scenarios. The benchmark would be strengthened by incorporating more tasks, including more complex ones that agents currently fail (as seen in the utility test), to better quantify the degree of "security by incompetence" and to "future-proof" the benchmark against more capable future agents.*
>
> **Response**: Thank you for your feedback on the evaluation scope. We agree that broader coverage is important, and we want to clarify our reasoning and context for the current setup:
>
> - The two environments we selected allow us to test a wide range of attacker and user goals, and to test multi-step, end-to-end workflows in depth. In addition, we already include adversarial objectives spanning from security (exceeding authorized access) to social harms (harassment, amplifying political messages, etc). Our results are consistent across these scenarios in two different environments. The vulnerabilities and attack patterns we observe are not unique to GitLab or Reddit—they’re likely to show up in any environment where agents interact with untrusted user content.
>
> - Running large LLMs—especially agentic models—is extremely resource-intensive. The benchmarking community is moving toward smaller but more diverse benchmarks (see arXiv:2402.14992). We followed this trend to keep the time and cost manageable: for example, running Claude-cua on our full benchmark takes about 72 hours and costs ~$1,500. So scaling this up to thousands of instances would be introducing a lot of extra cost with marginal benefit to generalizability.
>
> - Our benchmark size is 84 test cases, which is actually on par with several recent agentic benchmarks: AgentHarm (110 tasks), Tau-bench (165 tasks), etc. So, while the number of environments is limited, the overall task diversity and coverage are quite competitive.
>
> - The WASP framework is open-source and modular, so it’s straightforward to add new environments like e-commerce or travel booking in the future. We explicitly mention this as a direction for future work and welcome community contributions.
> In short, we focused on depth and realism within two environments to keep the benchmark practical and meaningful, while still offering broad task coverage. Expanding to more environments is definitely on our roadmap, and we see this as a community effort going forward.
>
> > The paper does not categorize the attacks based on the type of malicious behavior being attempted. For instance, attacks could be categorized as information theft, permission abuse (e.g., an agent using the user's credentials to perform unauthorized actions), or causing financial harm (e.g., making incorrect purchases). A taxonomy of attacks would be valuable for identifying which specific agent functionalities are most vulnerable and for designing more granular security controls.*
>
> **Response**: In our paper, we do provide such a categorization in the Appendix (see Tables 5 and 6), where each adversarial task is labeled according to its security violation category—such as information theft, permission abuse, data destruction, and harassment. We agree that further analysis according to the taxonomy would be interesting.
>
> > The core finding of "security by incompetence" highlights that the low end-to-end success rate is primarily a function of the agent's general capability, not its security robustness. While interesting, this means the security-specific insights are somewhat limited. The benchmark could be improved by designing tasks that better isolate security mechanisms from general task execution competence.*
>
> **Response**: As agents become more capable, we expect the gap between intent hijack and successful attack completion to narrow, making security-specific evaluation even more critical. Expanding the benchmark to include tasks and scenarios that directly test security mechanisms—independent of general competence—is a valuable direction for future work, and our framework is well-suited to support such extensions.
>
> In the meantime, we introduced the ASR-intermediate metric to captures whether an agent’s intent or reasoning was hijacked—even if the final action was safe—because it reveals underlying vulnerabilities in the agent’s decision-making process.
>
> > Some analyses in the paper could be deeper. For example, the o1 model had a surprisingly high ASR-intermediate despite its instruction hierarchy. The authors' initial analysis is appreciated, but further experiments would be good to root-cause this vulnerability and potentially flag a specific loophole in o1's security design.*
>
> **Response**: The reviewer raises an interesting point. We have conducted a deeper manual analysis of the traces from the agent backed by o1. We observe that the o1 model tends to complete both the adversarially injected task and the original agent user’s request. This likely points to an “instruction following bias” that might happen if the model is optimized to faithfully address all instructions in its context window. We would be happy to add this discussion to the paper in case of acceptance.
>
> Furthermore, we thank the reviewer for the suggestions for presentation and clarity and commit to implementing those in case of acceptance!
>
> > Analysis of Agent Reversion: Figure 3 shows some fascinating cases where the agent reverts to the original user's request after being hijacked. Could the authors investigate this further? It would be valuable to understand if this reversion is a sign of a latent security awareness (i.e., the agent realizes it was misled) or simply another form of incompetence where it fails to follow the attacker's instructions correctly.*
>
> **Response**: The observed agent reversion in Figure 3 is primarily due to the current design of the agentic scaffolding—in particular, the WebArena scaffolding always includes the original user instruction (and previous instructions) in the agent’s context at each step. This persistent presence of the user’s goal increases the likelihood that the agent will revert to the original request, even after being hijacked by an adversarial prompt. The reviewer is right to highlight that this more constrained scaffolding presents a form of a "natural defense mechanism" and makes the task of the attacker harder. This is similar to the motivations for developing the URL hijacking attacks and we would be happy to highlight this in the discussion in case of acceptance!

---

> > ### Comment · Reviewer_RYwN · 2025-08-05
> >
> > Thanks for the response. Most of my points were addressed. That being said, I'm still not quite convinced about the first point (limitation on diversity of test scenarios).
> >
> > > The vulnerabilities and attack patterns we observe are not unique to GitLab or Reddit—they’re likely to show up in any environment where agents interact with untrusted user content.
> >
> > My view is that one important part of evaluation benchmarks is to try to cover more diverse situations, even if the different tasks will "likely" show the same vulnerabilities and attack patterns. Of course, we also want patterns themselves to be diverse as well. Overall, the optimal benchmark should be diverse across multiple different criteria. I can understand though if it's not feasible for the authors for now, due to time and cost reasons.
> >
> > Looking forward to see more blocks added into the benchmark in the future.
> >
> > I've already gave an acceptance rating at the beginning and I'm still happy to accept the paper. I'm keeping the current score.

---

> > > ### Author Response · Authors · 2025-08-06
> > >
> > > Thank you for the thoughtful review! We agree and plan to extend the benchmark in the future and invite community contributions!

---

### Official Review · Reviewer_CzjF · 2025-07-01

**Rating:** 4
**Confidence:** 4

**Summary:**

This paper proposes WASP, a web agent security benchmark for evaluating prompt injection attacks. Unlike prior work, WASP emphasizes a more realistic setting by implementing template-based plain text and URL injections, and assesses the robustness of 11 leading AI models within the VisualWebArena platform. The results show that, while it is easy to distract agents, successfully hijacking them to accomplish an adversary’s ultimate goal remains very difficult—a phenomenon the authors term "security by incompetence." The paper also discusses potential mitigation strategies, including instruction hierarchies and defensive system prompting.

**Additional Feedback:**

None.

**Dataset Code Accessibility:**

Yes

**Ethical Comments:**

No obvious ethical concerns were identified, as all experiments were conducted in an isolated sandbox environment.

**Ethical Considerations:**

No, there are no or only very minor ethics concerns

**Final Justification:**

The rebuttal has addressed my concerns, so I are willing to increase the rating to 4.

**Limitations Weaknesses:**

1. The study evaluates only one web agent framework with a limited set of tasks and goals. In contrast, AgentDojo tests 97 realistic tasks (albeit for general agents). Did AgentDojo include web agents? How does this work advance beyond existing general and web agent benchmarks in terms of attack types, adversarial tasks, and application scenarios?

2. Only two types of manual prompt injection attacks are tested. A more comprehensive evaluation covering a wider variety of attack strategies is needed. The adversarial tasks should also vary in complexity and encompass a broader range of threats, such as privacy breaches, incorrect actions, operational errors, and misinformation retrieval.

3. The results suggest o1 is more vulnerable to end-to-end attacks compared to 40/min or Sonnet. What accounts for this increased vulnerability?

4. Lines 255–256 are unclear: “it does not stand to reason that the entire project needs to be deleted in order to comment.” Please clarify this statement.

5. The paper claims that “attackers are bottlenecked by agent capabilities,” yet these agents demonstrate strong normal performance and should be capable of executing adversarial tasks as well. Is this limitation due to the simplicity of the template-based attacks?

**Strengths Contributions:**

1. The proposed benchmark addresses an important and timely topic in agent safety.

2. The threat model reflects a more realistic scenario compared to previous studies.

3. The work provides in-depth empirical insights into both the effectiveness and challenges of hijacking web agents.

---

> ### Author Rebuttal · Authors · 2025-07-30
>
> We thank the reviewer for their valuable time reviewing the paper and their thoughtful comments! Here are responses to each point:
>
> > The study evaluates only one web agent framework with a limited set of tasks and goals. In contrast, AgentDojo tests 97 realistic tasks (albeit for general agents). Did AgentDojo include web agents? How does this work advance beyond existing general and web agent benchmarks in terms of attack types, adversarial tasks, and application scenarios?*
>
> **Response**: AgentDojo operates in an environment where an LLM simulates responses to tool calls and does not include web agents (only 4 task suites based on API interactions). By contrast, our benchmark is built on fully operational, self-hosted web platforms seeded with real data that closely mirror real-world usage and threats. The security of web and API agents is inherently different: While agents relying on APIs are restricted by the design of the API, web agents come with much more permission by default and no fine-grained access controls. For example, functions to manage the keys used to access a GitLab repository are not be available to an API agent such as those in AgentDojo but web agents by default come with the full permissions of a logged-in user of a website and can take more destructive/impactful actions. Additionally, adversaries face different restrictions in API and in web agents. Some web agents operate only with a screenshot view and do not take in a text representation of the page. This means adversaries are limited in the length of their injected prompt. By contrast, an email with an injection in AgentDojo can be arbitrarily long. We believe it is important for the community to be able to study attacks on realistic agentic workflows both on the web and in an API setting. That is why WASP intends to set a level playing field for attackers and modeling/defender teams under standardized and realistic threat models on the web. By focusing on realistic web environments, we are able to evaluate web agents under authentic, multi-step workflows and adversarial conditions that agents are likely to encounter in practice.
>
> In addition, we evaluate 4 different types of web agents: fully text-based web agents as proposed in the original WebArena paper (Zhou et al., 2023), set-of-marks agents from VisualWebArena (Koh et al, 2024), computer use agents (Anthropic, 2024), and a modernized version of the WebArena text-based agents making use of more recent tool-calling APIs. See Section 4.1 for an in-depth discussion of each of these agentic frameworks/scaffoldings.
>
> Running large WebAgentic models on such a realistic environment is extremely resource-intensive. For example, running Claude-cua on our full benchmark takes about 72 hours and costs ~$1,500. In addition, we already include adversarial objectives spanning from security (exceeding authorized access) to social harms (harassment, amplifying political messages, etc). Our results are consistent across these scenarios in two different environments.  So scaling this up to thousands of instances would be introducing a lot of extra cost with marginal benefit to generalizability.
> Additionally, the WASP framework is open-source and modular, so it’s straightforward to add new environments like e-commerce or travel booking in the future. We explicitly mention this as a direction for future work and welcome community contributions. As for the size of tasks and goals, our benchmark comprised of 21 attacker goals x 2 user goals x 2 injection mechanism = 84 unique test cases, which is actually on par with several recent agentic benchmarks: AgentHarm (110 tasks), Tau-bench (165 tasks), etc. So, while the number of environments is limited, the overall task diversity and coverage are quite competitive.
>
> References:
>
> Zhou, Shuyan, et al. "Webarena: A realistic web environment for building autonomous agents." arXiv preprint arXiv:2307.13854 (2023).
>
> Koh, Jing Yu, et al. "Visualwebarena: Evaluating multimodal agents on realistic visual web tasks." arXiv preprint arXiv:2401.13649 (2024).
>
> Anthropic. Introducing computer use, a new claude 3.5 sonnet, and claude 3.5 haiku. https://www.anthropic.com/news/3-5-models-and-computer-use, Oct 2024. Accessed: 03 February 2025
>
>
> > Only two types of manual prompt injection attacks are tested. A more comprehensive evaluation covering a wider variety of attack strategies is needed. The adversarial tasks should also vary in complexity and encompass a broader range of threats, such as privacy breaches, incorrect actions, operational errors, and misinformation retrieval.*
>
> **Response**: In this initial release, we focused on two main manual prompt injection strategies (plain-text and URL-based) to establish a clear, reproducible baseline and keep resource costs manageable. Within these, we cover a wide range of adversarial goals—like privacy breaches, privilege escalation, misinformation, and operational errors. Importantly, our results show that even with these relatively simple manual prompts, state-of-the-art agents are still highly susceptible to injection attacks, highlighting the urgency of addressing these vulnerabilities before moving on to even more sophisticated adversarial strategies. That said, expanding to more attack types and complexities is a key direction for future work, and our framework is designed to make this easy.
>
> > The results suggest o1 is more vulnerable to end-to-end attacks compared to 40/min or Sonnet. What accounts for this increased vulnerability?*
>
> **Response**: The o1 model appears more vulnerable to end-to-end attacks because its advanced reasoning and action capabilities make it more likely to follow through on complex adversarial instructions once hijacked. In contrast, less capable agents often get confused or fail to complete multi-step attacks. This highlights a key risk: as agents get smarter, their security weaknesses can become more dangerous if not addressed.
> We also spot-checked 42 execution traces and the corresponding ASR-intermediate labeling by the judge by manually annotating the actions without looking at the judge label. We found 7 false positives and 0 false negatives in this set. We believe that any bias in the judge, however, is consistent across models. Finally, ASR-end-to-end results are not probabilistic and they rank the models in the same way, so we have confidence in our overall reported numbers.
>
> > Lines 255–256 are unclear: “it does not stand to reason that the entire project needs to be deleted in order to comment.” Please clarify this statement.*
>
> **Response**: That line means that some adversarial prompts instruct the agent to do things that are clearly unnecessary or excessive—like deleting an entire project just to add a comment. The point is to show that agents can be misled by implausible instructions, which underscores the need for better safeguards against prompt injection.
>
> > The paper claims that “attackers are bottlenecked by agent capabilities,” yet these agents demonstrate strong normal performance and should be capable of executing adversarial tasks as well. Is this limitation due to the simplicity of the template-based attacks?*
>
> **Response**: Thank you for this observation! Our empirical findings actually point to the opposite: agents struggle with completing the end-to-end tasks of moderate complexity. Consider the Utility column in Table 2: almost none of the tested agents can complete more than half of tasks that the legitimate user of the agent requests with no adversary present. Additionally, we hand-labeled some trajectories of agents under attack to better understand attack failure modes. Those results are shown in Figure 3. Note that in most cases an agent does not recover after being hijacked but rather simply fails to complete the task set by the adversary (the red set of states on the left flows to mostly red states to the right). This points to the fact that our attacks are successful at getting the agents to adopt the adversarial goals but the utility of the agents to complete them is limited.

---

> > ### Comment · Reviewer_CzjF · 2025-08-03
> > **Thanks for the clarifications.**
> >
> > The rebuttal has addressed my concerns, so I am willing to increase the rating to 4.

---

> > > ### Author Response · Authors · 2025-08-06
> > >
> > > Thank you! We appreciate the feedback!

---

### Official Review · Reviewer_i9jW · 2025-07-02

**Ethics Flags:** Safety and security
**Rating:** 5
**Confidence:** 3

**Summary:**

This paper presents WASP, a benchmark designed to assess the security of autonomous web navigation agents against prompt injection attacks. Unlike many prior benchmarks that assume unrealistic attacker capabilities or focus on isolated tasks, WASP provides a more realistic setup: the attacker is modeled as an adversarial user with limited control over web content (e.g., comments, posts) rather than the entire environment. The benchmark supports end-to-end testing in sandboxed GitLab and Reddit clones and evaluates the susceptibility of agents to prompt injection in these environments.

**Dataset Code Accessibility:**

Yes

**Dataset Code Comments:**

The dataset are well presented and easily accessed.

**Ethical Comments:**

The evaluation should be under well-controlled environment.

**Ethical Considerations:**

Yes, there are ethics concerns that require attention by the authors

**Final Justification:**

Thanks for the response. I raise my score to 'accept'.

**Limitations Weaknesses:**

1. Limited evaluation scope and diversity: The evaluation is primarily restricted to two environments (GitLab and Reddit clones). While the authors acknowledge this limitation, the benchmark would benefit significantly from a broader set of environments (e.g., e-commerce, knowledge bases, travel booking platforms). As it stands, it is unclear whether the results generalize beyond these narrow settings.
2. ASR focus on objective: The ASR-intermediate metric appears to focus on the intent or objective behind agent actions (as judged by LLM analysis), even in cases where the actual execution behavior may have been safe. Why is it important if the behavior was ultimately safe?

**Strengths Contributions:**

1. The paper tackles an important and timely problem, the vulnerability of web agents to prompt injection, which is critical as such agents are being deployed for real-world tasks (e.g., browsing, shopping, filling forms).
2. The benchmark realistically models the attacker’s capabilities, avoiding assumptions like full control of the external environment or insider access to the agent’s inner state.
3. It is good that the paper also consider the evaluation under multiple defense strategies.
4. The benchmark is publicly available, promoting transparency and reproducibility, which is highly valuable for the research community.

---

> ### Author Rebuttal · Authors · 2025-07-30
>
> We thank the reviewer for their engagement with this work and the thoughtful comments! Here are our responses to each point:
>
>
> > Limited evaluation scope and diversity: The evaluation is primarily restricted to two environments (GitLab and Reddit clones). While the authors acknowledge this limitation, the benchmark would benefit significantly from a broader set of environments (e.g., e-commerce, knowledge bases, travel booking platforms). As it stands, it is unclear whether the results generalize beyond these narrow settings.*
>
> **Response**: Thank you for your feedback on the evaluation scope. We agree that broader coverage is important, and we want to clarify our reasoning and context for the current setup:
> - The two environments we selected allow us to test a wide range of attacker and user goals, and to test multi-step, end-to-end workflows in depth. In addition, we already include adversarial objectives spanning from security (exceeding authorized access) to social harms (harassment, amplifying political messages, etc). Our results are consistent across these scenarios in two different environments. The vulnerabilities and attack patterns we observe are not unique to GitLab or Reddit—they’re likely to show up in any environment where agents interact with untrusted user content.
> - Running large LLMs—especially agentic models—is extremely resource-intensive. The benchmarking community is moving toward smaller but more diverse benchmarks (see [Polo et al. 2024 tinyBenchmarks: evaluating LLMs with fewer examples](https://arxiv.org/abs/2402.14992)). We followed this trend to keep the time and cost manageable: for example, running Claude-cua on our full benchmark takes about 72 hours and costs ~$1,500. So scaling this up to thousands of instances would be introducing a lot of extra cost with marginal benefit to generalizability.
> - Our benchmark size is 84 test cases, which is actually on par with several recent agentic benchmarks: AgentHarm (110 tasks), Tau-bench (165 tasks), etc. So, while the number of environments is limited, the overall task diversity and coverage are quite competitive.
> - The WASP framework is open-source and modular, so it’s straightforward to add new environments like e-commerce or travel booking in the future. We explicitly mention this as a direction for future work and welcome community contributions.
> In short, we focused on depth and realism within two environments to keep the benchmark practical and meaningful, while still offering broad task coverage. Expanding to more environments is definitely on our roadmap, and we see this as a community effort going forward.
>
>
>
> > ASR focus on objective: The ASR-intermediate metric appears to focus on the intent or objective behind agent actions (as judged by LLM analysis), even in cases where the actual execution behavior may have been safe. Why is it important if the behavior was ultimately safe?*
>
> **Response**: It’s important to measure ASR-intermediate, which captures whether an agent’s intent or reasoning was hijacked—even if the final action was safe—because it reveals underlying vulnerabilities in the agent’s decision-making process. Our results show that many agents are only “safe” because they fail to fully execute complex tasks, a phenomenon we describe as “security through incompetence.” If an agent is consistently diverted from its original objective by adversarial prompts, it signals that the agent is susceptible to manipulation, and as agent capabilities improve, these currently “safe” failures could easily become real security incidents. By identifying when the agent’s reasoning is compromised, we can proactively address weaknesses before they result in harmful or unsafe behaviors as agents become more competent and autonomous.
>
>
> > Ethical Considerations: Yes, there are ethics concerns that require attention by the authors*
>
> > Ethics Flags: Safety and security*
>
> > Ethical Comments:*
>
> > The evaluation should be under well-controlled environment.*
>
> **Response**: Thank you for raising this point. All of our evaluations were conducted in fully self-hosted, sandboxed environments that are isolated from real users and external systems. This setup ensures that no real-world harm can occur and that all security and safety testing is performed under strictly controlled conditions. We believe this approach aligns with best practices for responsible AI research and evaluation.

---

> > ### Comment · Reviewer_i9jW · 2025-08-04
> >
> > Thanks for the response. I raise my score to 'accept'.

---

> > > ### Author Response · Authors · 2025-08-06
> > >
> > > Thank you so much! We appreciate the thoughtful engagement!

---

### Comment · Area_Chair_1kUi · 2025-08-08

Dear Authors and Reviewers,

The deadline for author-reviewer discussions is approaching. If you haven’t done so already, please ensure there are sufficient discussions for both the submission and the rebuttal.

Reviewers, please make sure you complete the mandatory acknowledgment AND respond to the authors’ rebuttal, as requested in the email from the program chairs.

Authors, if you feel that any results need to be discussed and clarified, please notify the reviewer. Be concise about the issue you want to discuss.

Your AC

---

### Decision · Program_Chairs · 2025-09-18

**Decision:**

Accept (poster)

**Comment:**

This paper introduces the WASP benchmark, designed to evaluate the sensitivity and robustness of Web Agents against prompt injection in realistic scenarios. It identifies the “Security by incompetence” phenomenon and discusses potential mitigation strategies. The contributions are clear.

All reviewers have provided positive scores for this submission, highlighting its strengths and key contributions. Though they also raise concerns consistent with my own observations. In particular, the experiments cover only two environments with a total of 84 tasks, which may limit the generalizability of the conclusions; models specifically fine-tuned for Web Navigation were not evaluated. These weaknesses are also clear and limit the contribution of this paper to the community.

After careful review and discussion, the AC thinks the methodology is overall sound and the analysis is valuable, supporting the decision to accept this submission. However, the AC also suggests that the authors should expand the experimental coverage in the final revision.